# Biomimetic Origami: A Biological Influence in Design

**DOI:** 10.3390/biomimetics9100600

**Published:** 2024-10-04

**Authors:** Hadi Ebrahimi Fakhari, Juan Rosario Barboza, Pezhman Mardanpour

**Affiliations:** Department of Mechanical and Material Engineering, Florida International University, 10555 W Flagler St, Suite 3464, Miami, FL 33174, USA; hebra007@fiu.edu (H.E.F.); jrosa173@fiu.edu (J.R.B.)

**Keywords:** origami-inspired structures, biomimetic origami, taxonomy, DNA origami, robotics

## Abstract

Origami, the art of paper folding, has long fascinated researchers and designers in its potential to replicate and tap the complexity of nature. In this paper, we pursue the crossing of origami engineering structures and biology, the realm of biologically-inspired origami structures categorized by the two biggest taxonomy kingdoms and DNA origami. Given the diversity of life forms that Earth comprises, we pursue an analysis of biomimetic designs that resemble intricate patterns and functionalities occurring in nature. Our research begins by setting out a taxonomic framework for the classification of origami structures based on biologically important kingdoms. From each of these, we explore the engineering structures inspired by morphological features, behaviors, and ecological adaptations of organisms. We also discuss implications in realms such as sustainability, biomaterials development, and bioinspired robotics. Thus, by parlaying the principles found in nature’s design playbook through the art of folding, biologically inspired origami becomes fertile ground for interdisciplinary collaboration and creativity. Through this approach, we aim to inspire readers, researchers, and designers to embark on a journey of discovery in which the boundaries between art, science, and nature are blurred, providing a foundation for innovation to thrive.

## 1. Introduction

The paper-folding craft known as origami, derived from Japanese culture, has fascinated generations due to its simplicity, elegance, and transformative power; intricate forms arise with one sheet of paper, reflecting the creativity and ingenuity of human expression. Origami is a Japanese paper craft originally practiced for ceremonial and recreational purposes. It includes carefully folding a flat sheet of paper into a finished sculpture composed of a repeating pattern without cutting or adhesives. The discipline’s innate originality and precision have progressively expanded its applicability outside the creative sphere, where it has become highly valuable across various engineering areas. Yet, origami is embedded in a deeper connection to the world of nature, with the patterns of folding echoing the rhythms and geometries of life itself.

A noteworthy step in the transformation of origami from an art form to a structural tool occurred in the middle of the 20th century, as academics started to look at the possibilities of origami-inspired designs for producing effective, portable, and small structures. This shift in application started to garner a great deal of attention, with Japanese astronomer Dr. Koryo Miura making a groundbreaking contribution to the discipline in 1969 when he introduced the Miura fold. The Miura fold is a type of rigid origami that may be folded continuously so long as each parallelogram stays perfectly flat at every step. Notably, the Japanese space program has used the Miura fold for massive solar panel arrays on space satellites, which can be unfolded after being launched and placed in orbit. It is possible for a Miura fold to be compacted into a shape consisting solely of the thickness of the folded material. By pulling on the opposing ends of the cloth, it can be unpacked in a single action, then folded again in a similar manner by pushing the two ends together. This feature notably minimizes weight and complexity in the solar array application by lowering the number of motors needed to unfold the structure [1].

Interest in biomimicry in design has converged with origami to create the new frontier of biologically-inspired origami structures. As acknowledged by Rus and Tolley [2], a significant portion of modern origami robots are inspired by biomimicry. Drawing inspiration from the extraordinary diversity of life on Earth, researchers and designers have started to investigate how the principles of biology can inform the folding techniques and design paradigms of origami.

This paper embarks on a journey through the interdisciplinary realm of origami inspired by biology. It is guided by a taxonomic approach, organizing biologically-inspired origami structures according to two categories: first, the kingdoms of life, consisting of Animalia and Plantae; and second, DNA origami (Figure 1). A mosaic of folding patterns and design principles is seen within each of these categories, reflecting the morphological features, behaviors, and ecological adaptations of organisms. As described by Nojima et al. [3], living organisms can be used as a guide in research to develop shape-changing origami models.

Beyond mere imitation, biologically inspired origami holds promise for innovation and problem-solving in a wide variety of fields. From biomaterials to architecture and robotics to aerospace, the fusion of nature’s design principles with the art of folding opens doors to novel solutions and sustainable technologies; in addition, the study of biologically inspired origami contributes to fundamental questions about form, function, and evolution, enriching our understanding of both nature and design.

## 2. Origami Background

Origami is the art of folding paper, as mentioned by Debnath and Fei [4]; the term “origami” itself comes from the Japanese words “oru”, for “to fold”, and “kami”, which means paper. According to Koshiro Hatori [5], origami dates back to the Muromachi period (1333–1573) in Japan, where it was first documented as being used in ceremonies. Now, residents worldwide practice folding origami as a science and for recreation. Over the past few decades origami has sparked interest in artists and scientists alike, motivating them to study the principles underlying this art form. Scientific interest has shifted the origami art form to a mathematical discipline; hence, origami techniques are finding applications across industries, and are becoming useful in sectors such as robotics and biomimicry. Following the work of Koryo Miura [1], researchers have further evaluated the potential within origami structures. These structures have limitless capabilities to enhance design, including multi-stability, flexibility, reduced space, and reduced cost. Origami patterns exhibit a number of properties that make them relevant in numerous fields; for example, variable stiffness can be explained as the ability of materials to change their response to forces, while negative stiffness causes the material to deform more easily than it otherwise would. Multi-transformability in an origami structure refers to the structure’s ability to change from one shape or form to another. Meanwhile, multi-stability denotes the property of a system to maintain stability in a number of distinctive shapes or states. Several of the most relevant origami folding patterns that have been “re-discovered” and applied to engineering designs are discussed below.

### 2.1. Miura Fold

It is possible to create Miura’s pattern by tessellation of several components (Figure 2) [6]. Four identical parallelograms make up a unit. Because the Miura fold provides single-Degree Of Freedom (DOF) mobility, the unit’s motion solely depends on the dihedral angle, where the crease lengths and twist angle are specified as constants [6]. Because of the fold’s special design, structures can be packed and deployed quickly, allowing for seamless expansion and contraction. Because of this property, it is helpful for designing flexible, lightweight, and robust solutions in robotics, where weight and space restrictions are crucial. Furthermore, the Miura fold distributes stress evenly and offers great structural integrity, which improves the resulting constructions’ durability and resistance in a variety of environmental circumstances.

### 2.2. Kresling Fold

The Kresling pattern is made by folding a flat, rectangular sheet of material into a number of triangular facets that produce a structure similar to a helical prism when compressed. The helical folding patterns that Kresling structures exhibit provide considerable compressibility and expandability (Figure 3) [7]. The high strength-to-weight ratio offered by Kresling patterns make it possible to build sturdy and lightweight structures that are able to support heavy loads [7]. As described by Moshtaghzadeh et al. [8,9,10,11,12,13], an origami construction with stronger panels and flexible creases is more resistant to fatigue failure. A notable feature of Kresling patterns is their bistability, that is, the ability to remain stable in compressed and expanded states without requiring constant energy input. Furthermore, because of their special geometric characteristics, mechanical behaviors such as stiffness and flexibility can be adjusted and tailored to particular uses.

### 2.3. Waterbomb Fold

The classic waterbomb origami fold (Figure 4) [14] is created from a design made up of vertices in which six creases converge. It has several degrees of freedom from a rigid origami perspective, but just one when the design is folded symmetrically [14]. Because of the natural flexibility of the waterbomb fold, stresses may be equally absorbed and distributed, increasing the resulting constructions’ resilience and endurance. Its straightforward but efficient design can also be adjusted to suit different forms and sizes, offering a scalable way to make foldable, lightweight, and structurally sound components for both small-scale and large-scale engineering projects.

### 2.4. Flasher Fold

The flasher origami pattern, as discussed by Jang et al. (Figure 5) [15] is distinguished by its dynamic radial folding pattern, which allows for a special transition between two-dimensional (2D) and three-dimensional (3D) states. Its most remarkable characteristic is its capacity to stretch and contract outward from a central core, producing a flaring or “flashing” movement. This mechanism is perfect for applications that need to expand quickly, such as satellite structures or deployable shelters, as it can be used to create large-scale structures from a compact shape. Furthermore, the flasher design allows for seamless transitions between 2D and 3D shapes, offering flexibility for mechanical, architectural, and creative applications where adaptability and space efficiency are critical.

Origami is defined by its foundational creases that tell the paper where to bend into shape. These creases shape the final model, and are respectively called “mountains” or “valleys”. Origami’s dimensions refer to spatial relationships and proportions within the folded elements, which are of crucial importance in accurately replicating different patterns. This combination of geometrical precision with creative craft underlines the unique appeal of origami, providing a synthesis of mathematical rigor and artistic expression.

## 3. Biomimetic Origami

For simplicity, we investigate origami structures through the most prominent kingdoms of life, namely, Animalia and Plantae, as well as other worthy nature inspirations, in order to explore the rich designs of origami engineering structures that have been inspired by nature. Each fold and crease bears witness to the ingenious perfection that evolution has brought about in organisms over millennia. In the following pages, we find not just the beauty of biologically inspired origami but also its practical applications in engineering and design. From the streamlined contours of animal anatomy to the fractal geometries of plant leaves, including both larger macrostructures and tiny nanostructures, we uncover inspiration for engineering innovation.

### 3.1. Animalia-Inspired Origami

In this section, we discuss how origami engineering has been inspired by the incredibly unique organisms seen in nature. We review the biomechanics and structural adaptations of crawling species, nimble insects, marine creatures, and other members of the Animalia kingdom. Through meticulous observation and in-depth study, it has been possible to identify the design principles that allow animals to move freely in various settings, manipulate things, and display seemingly unattainable levels of power and agility. This creates opportunities for creative technical solutions that mimic the effectiveness and grace of natural designs.

Interesting results can be obtained by exploring the overlap between different fields; Matthew Gardiner [16] talked about the intersection of origami and robotics, called oribotics, and provided a number of subjects to build it up. Important subjects include oribotic origami pattern design, the incorporation of biomimetic crab joints into exoskeletons, and metaphorical depictions of the biological interaction of foraging bees, all of which encourage admiration for folded forms of nature and help to incorporate them into man-made settings.

#### 3.1.1. Crawling Robots

The construction of an origami-based robot that moves in a similar way to an earthworm was shown by Fang et al. [17]. The robot is inspired by the softness of origami and the morphology of earthworms. The robot’s mobility mimics that of a worm by employing origami ball constructions with different actuation methods, similar to how earthworms move. In this way, the earthworm body segments’ bidirectional deformations are replicated. Experiments demonstrated that the origami ball’s compliance and multi-stability are advantageous for the construction of robots. Three distinct robot segments were designed utilizing various actuators and an origami ball, while a six-segmented robot prototype was built using a DC motor-driven segment. The created prototype exhibited good locomotion, with various modes and speeds controlled by a gait generator modeled after earthworm movement. Future origami-based robot development appears to have a bright future thanks to its low cost of production, customizability, and scalability (Figure 6) [17].

Jin et al. [18] designed an origami robot inspired by worm biomechanics that uses a paper-knitting method to perform complex movements and tasks. The backbone of the robot can flex and deform under tension, compression, and bending, and it is propelled by magnetic forces and torques. The robot can clear hurdles, scale walls, and move freight, as shown in their experiment’s numerical simulations and theoretical analysis. The robot is low-cost, lightweight, and flexible, and can be used in various environments. Ze et al. [19] presented a “small-scale origami crawler” based on a Kresling dipole inspired by the crawling motion of earthworms. This magnetically operated crawler is proposed for biomedical application, and consists of a four-unit Kresling origami assembly that can navigate in restricted spaces. While Ze et al. aimed for smaller versions of their Kresling crawler, it would also be interesting consideration to consider a larger scale and its implications. Meanwhile, Wei et al. [20] addressed the limitations of soft robotics by increasing the bending angle, elongation rate, and movement flexibility of a tubiform origami structure inspired by annelid somite joints. They used a Ni–Ti memory alloy wire for actuation by eliminating negative Poisson ratio and removing material, which enabled efficient and flexible extension (Figure 7) [20].

A biomimetic origami prototype robot in the shape of a caterpillar was presented by Cat et al. [21], consistsing of a biomimetic origami robot that can move independently in tight spaces. This robot is highly adaptable, with a the creep pattern that can be controlled to enhance mobility. It can perform compression, deformation, six modes of locomotion, and various motion types with good stability, tunable elasticity, and adaptive locomotion. This adaptability makes the robot particularly suitable for complex environments and precise tasks. Similarly, Xu et al. [22] expanded the field of Kresling origami by mimicking a worm, with motion originating through vacuum chambers that are capable of achieving unidirectional displacement under tight conditions.

Similarly, the Origami Robotic Snake (OriSnake) by Luo et al. [23] is a soft robotic platform for search-and-rescue applications. Unlike traditional snake robots that use links and joints, OriSnake uses tubular origami continuous sectors driven by inner cords and electric mechanisms for distributed actuation, sensing, and control. Modeled after real snake movements, OriSnake’s lateral undulation and sidewinding gait allow it to move in 3D spaces. Experimental results show that OriSnake can move at 40.5 mm/s in lateral undulation gait and 35 mm/s in sidewinding gait. Its light weight, low cost, and origami-based flexibility enable it to navigate through narrow spaces and 3D terrain (Figure 8) [23]. Zuo et al. [24] presented another snake-shaped robot; inspired by waterbomb origami concepts, their design includes modular joints with flexible hinges and soft rubber material, and can bend and deform through lasso constructions. Kinematics modeling and control system implementation allow it to move on various surfaces.

Many different authors from various backgrounds have presented aligned ideas with a range of similarities; the majority of crawlers we reviewed use the waterbomb pattern as a building block for robots, as this pattern provides the full range of desired motions while distributing the stress loads towards the confined spaces required for crawler robots. On the other hand, the Kresling and Yoshimura patterns are used only for extremely specialized areas, in particular those requiring empty inner cavity spaces and particular ranges of motion, respectively. Ultimately, crawling robots need further improvement; fully untethered capabilities and high range of motion are among the development areas where researchers are aiming to find solutions.

#### 3.1.2. Dual Morphing

Mintchev et al. [25] described a creative kind of origami construction modeled after insect wings. This construction exemplifies dual stiffness by fusing resilient and soft materials with the supporting strength of rigid materials and the potential to change shape. The origami structure serves as the frame for a gripper quadcopter, which operates in an aerodynamic environment and has the ability to change shape to prevent damage in the event of a collision. This property makes it possible to integrate electronic circuitry and soft actuators into the origami to provide proprioception and controlled actuation. This design opens up multipurpose shape-shifting structures for consumer applications as well as robotics and aerospace, such as morphing drone wings with distributed sensors to improve flight stability.

Intricate constructions such as those found in insect wings have inspired a number of innovative structural designs. Baek et al. [26] examined the formation of compliant origami modules with strong “self-locking” capabilities that were modeled after the veins in ladybird beetle wings. These structures, which stray from traditional origami design, are able to support high loads and can be stored or deployed. This research examines how compliant origami can be used to improve the energy storage of a jumping robot’s mechanism and create a glider module that can be deployed for a “multimodal robot”. Additionally, Baek reviewed the use of compliant origami in intricate origami constructions with an emphasis on robotic systems applications. (Figure 9) [26]. However, one cause for concern that has been brought into discussion involves the activation of the gliding mechanism, as the entire system can be compromised if it fails.

The earwig wing folding mechanism has features that cannot be replicated by traditional origami; according to Faber et al. [27], these are rapid self-folding movements using a characteristic Resilin material, a wing pattern with angled and curved creases, and stability mechanisms that keep the wings latched during flight. Inspired by this biological mechanism, the authors proposed a spring origami model that expands the scope of traditional origami (Figure 10) [27]. Similarly inspired by earwig wings, Rojas et al. [28] proposed a technique using double-layered creases that contract asymmetrically in response to external input. They fabricated biomimetic earwig wings inspired by Miura’s fold and developed an analytical model to predict folding angles. They demonstrated their method on various membrane-based structures such as solar arrays through experimentation and finite element analysis, and discussed the possibility of creating self-locking mechanisms to stiffen biologically inspired origami structures. While Rojas et al. approached the challenge of mimicking an earwig’s wing and came up with relevant contributions regarding material selection and behavior, a gliding test is encouraged in order to enhance this contribution.

Saito et al. [29] provided guidelines for biomimicry and proposed a method to create deployable structures inspired by earwig wing folding. They used X-ray microcomputed tomography to reconstruct the geometrical rules of earwig wing crease patterns. This method was applied to design artificial deployable structures for mechanical engineering, aerospace, architecture, and everyday objects. While Faber et al. [27] and Rojas et al. [28] focused on the mechanical and adaptive aspects of earwig folding, Saito et al. [29] focused on the geometrical reconstruction and practical implementation of these principles in various fields and showed a more comprehensive range of applications for biomimetic deployable structures. Both approaches face unforeseen challenges and need additional research before reconstructing a synthetic earwig-inspired wing becomes possible.

Biological inspiration may be found in every living being; Kim et al. [30] examined the pelican eel’s unusual “dual-mode” morphing, which increases the chance of engulfing prey by first unfolding and then inflating the mouth. Similar to the earwig wing, this type of morphing is not achievable with traditional single-mode morphing mechanisms. The authors suggested dual-morphing structures that consider origami unfolding and skin stretching due to fluid pressure. They demonstrated quasi-sequential dual-morphing responses in artificial organisms for adaptive grabbing, crawling, and submerged mobility using only stretchable origami pieces. The fundamental building block of this new paradigm for constructing adaptive “shape-morphing systems” in soft robotics, engineering structures, and active metamaterials is the stretching of Miura and Yoshimura origami unit cells. (Figure 11) [30]. The double-folding nature of Miura’s fold and the stiffness required to both remain straight and stretch provided by Yoshimura designs provides solutions based on origami patterns to address the challenges of deploying pelican eel-inspired biomimicry.

#### 3.1.3. Inflatable Motion

Clarice et al. [31] presented PuffBot, an inflatable origami robot with movement and inflationary properties are modeled after those of a pufferfish. PuffBot uses a chemical process to inflate an internal balloon beneath its origami exoskeleton, which enables it to move in water through surge, yaw, and heave motion. Utilizing a microcontroller, a solenoid valve controls the inflation process, which can be managed remotely. Tests carried out for design verification demonstrated that PuffBot can successfully mimic a pufferfish. In addition to the conducted research, it would be ideal to test the reaction of this inflatable robot in different water bodies as well as its fatigue life when exposed to prolonged submersion.

Similarly, marine biology and origami have inspired soft robotic jellyfish. Hu et al. [32] used mathematical modeling and hydrodynamic simulation to select an origami polyhedra model that mimics the jet propulsion of prolate medusae. Their rope-motor-driven robotic jellyfish prototype can swim effectively, showing the system’s scalability, simplicity of fabrication, and structural efficiency (Figure 12) [32]. Additionally, in 2024 [33] Hu and Li expanded the capabilities of a “Rhombic Dodecahedron” to provide an inflatable origami concept. The goal was to present a compact and modular design with adaptability to varying environmental conditions. Qiu et al. [34] designed a biomimetic process for jellyfish origami using waterbomb tessellations. They explored geometric models and kinematic equations of the Biomimetic Jellyfish Origami Mechanism (BJOM) and optimized it for material usage, fineness ratio, and volume ratio. The optimized BJOM prototype showed better volume and fineness ratio and was able to mimic the flexible propulsion modes of natural jellyfish movement. This design could have many applications in underwater vehicles due to its effective propulsion mechanism and environmental adaptability (Figure 13) [34]. Though inspired by different biological systems and applications, both of these papers show the potential of biomimicry and origami in developing new robotic systems. The origami patterns selected by the authors grant these robots the ability to perform required functions without exerting unbearable stress on the structures, ideally while also mimicking biological organisms.

#### 3.1.4. Mechanical Transitions

Daynes et al. [35] introduced a structural idea modeled after adaptable morphing cells. A regulated bistable mechanism changes the configuration from flat to textured. Based on the concepts of kirigami, it uses silicone rubber cells strengthened by locally reinforced regions. Pneumatic actuation is used to fold or unfold the cells, while structural bistability ensures that the shape is maintained without the need for mechanisms or prolonged actuation. Finite element evaluations regarding shape and stability during actuation and a mathematical quantification of surface roughness were conducted. The authors emphasized the significance of angular rotation with respect to the deployment angle (Figure 14) [35]. Additional ways to improve this research could involve providing alternatives for non-flat surfaces and verifying their outcomes.

The shrimp origami pattern presented by Liu et al. [36] allows for the creation of programmable metastable phases through mechanical phase transitions. The two-dimensional units that make up the shrimp pattern can be tessellated in either homogeneous or heterogeneous local geometries, making it possible to design intricate energy landscapes inside the metamaterial. As a result, the metamaterial can change its mechanical characteristics and structure, which is useful for applications such as energy storage systems, reconfigurable acoustic waveguides, microelectromechanical systems, and reprogrammable materials of various scales (Figure 15) [36].

The biomimetic origami robot called Pillbot, developed by Zhang et al. [37], is based on the movement of pill bugs. This robot has three bionic elements: bionic pereiopods for movement in harsh conditions, soft internal muscles for functionality, and a flexible origami exoskeleton to reduce friction. The paper shows how soft robotics can use nature-inspired origami concepts to make robots more durable and usable, providing a practical example of biomimetic design in robotics. Its flat-bed origami design allows the structure to deploy, and its dynamic characteristics provide an example of how origami can impact engineering. Similarly, Baruah et al. [38] drew inspiration from the biomechanics of pill bugs to propose new engineering structures. Their research involved using origami-inspired designs for adaptive civil engineering applications, for which they analyzed the shape-changing properties of pill bugs. They compared a computer vision method applied to a 3D printed model with a dynamic relaxation technique for quasi-static form-finding of the origami pill bug structure. Experimental validation showed the efficiency of the dynamic relaxation module. In contrast, computer vision-based tracking showed the kinematic properties and dynamic behavior of the structure during rolling (Figure 16) [38]. These papers show the possibilities of pill bug-inspired origami in different fields: Zhang et al. [37] focused on applications in robotics, in particular integrating bionic elements to provide movement and durability in various conditions; in contrast Baruah et al. [38] explored the possibilities in civil engineering, using advanced analysis to understand and replicate the shape-changing properties of pill bugs for adaptive structures. Although they focus on different applications, both papers show the versatility and innovation of biomimetic origami in engineering and robotics.

Kamrava et al. [39] designed a mechanical string inspired by origami that can fold and position itself precisely and in specific forms with kinetic and kinematic motion based on programming. This system is actuated mechanically along a single line and has one degree of freedom, allowing it to be controlled from one end while keeping the exact position along its length. Applications include a biomimetic hand and a robotic gripper, both of which require high dexterity and sensitivity. The design is scalable across different length scales, making it suitable for scaled applications (Figure 17) [39]. However, it one cause for concern is how the gripper yields as the scale increases or decreases. In contrast, Green et al. [40] used biomimicry in regenerative ophthalmology to create biomaterials for treating blindness and visual impairment. By mimicking extracellular matrix features such as topography, biomaterials can promote cell organization and tissue regeneration. Their study presented two cases: the regrowth of ocular epithelium on nanostructured insect wing surfaces, and origami-inspired membranes for ocular cell transplantation. These biomimetic materials are designed to replicate natural basement membranes to solve regenerative ophthalmology challenges by providing clinically relevant solutions through advanced materials chemistry and self-deploying membranes. While tissue regeneration is a novel topic and further research is required, defining structures that could mimic the building blocks at a cellular level can open future research opportunities in the field. Similarly, Song’s work [41] used biomimetic design to create a compliant structural skin for biomimetic robots inspired by natural epithelium structures. The skin comprises rigid iron rings between a soft polyester fabric, and can stretch and bend with minimal resistance or energy consumption. Experimental results showed that the skin works underwater and aerially, has low cost, is easy to fabricate, and can be applied to various biologically-inspired structures such as worms, snakes, and fish robots. The flexibility and folding ratio can be adjusted by changing the number of rings, making it suitable for various robotic designs.

Kamrava et al. [39] and Song [41] both used biomimetic structures in robotics; Kamrava focused on mechanical actuation and control for precise movement, while Song focused on compliant materials for flexibility and adaptability in different environments. Green et al. [40] used biomimicry in medical biomaterials, specifically in regenerative ophthalmology. Each of these studies shows that biomimetic designs can be achieved and applied to different challenges, as different origami patterns behave in multiple predictable paths when the pattern tessellation is defined.

Seo et al. [42] used a fabric-integrated actuation module with an origami design to increase the strength and supporting force of soft pneumatic actuators. In contrast to conventional paper-based origami structures, this module features a stiff fabric origami pattern that uses less air and requires more effort. Specifically, the actuation module for the Yoshimura pattern was mathematically analyzed to adjust all design parameters in order to support the upper human limb. The module was experimentally validated lifting up to 7.5 kg at a pressure of 50 kPa or less, which is suitable for helping a person move their arm while holding a tool. The module was intended as the basis for a wearable system that would convert natural arm motion for overhead jobs (Figure 18) [42]. While the research was conducted and inspired by human limbs, it could also be tested beyond these limits to enhance human capabilities through robotics engineering that could take advantage of optimal origami design.

#### 3.1.5. Grippers

Nguyen et al. [43] provided an extensive overview of bioinspired grippers with parameters related to operating principles, materials, actuation, design complexity, fabrication techniques, and applications. Using biologically inspired designs, their review covered advancements in gripper technology, including both unyielding and soft gripper systems. The latter are inspired by living entities which work well in delicate and complex activities, while the former mimic the limbs of humans and animals to increase productivity in industrial automation and manipulation processes. Through their synergistic influence, biology and robotics have combined to create stunning gripper designs with applications across various industries. For example, Liu et al. [44] presented a new origami chomper-based flexible gripper design combining origami and a newly developed nonlinear topology optimization (NTO) method. The gripper performed well in various experiments, including gripping range under the same load, maximum gripping ratio, adaptability to different object textures and shapes, and scalability from millimeters to decimeters. By optimizing the origami structure with the NTO method, their gripper showed better gripping effectiveness and adaptability to irregular objects. In addition, their study demonstrates the computational efficiency and design refinement of the NTO method and its application in designing high-performance flexible grippers to handle objects with different stiffness, shape, size, and orientation. In summary, this work provides a foundation for designing new flexible grippers by combining simple origami with advanced optimization techniques to improve gripping and versatility across scales. Irrespective of the particular origami pattern or what different authors have achieved through their use, it is notable that the versatility of origami-inspired structures is inspiring developments in a wide variety of engineering fields.

#### 3.1.6. Other Applications

Zhang et al. [45] presented a novel approach to the production of reprocessable and programmable elastomeric sheets that are selectively altered by solvent-containing active particles and laser ablation. For soft ferromagnetic origami robots, this opens up a variety of functionalities including actuation, sensing, and adaptive coloration. The suggested method provides versatility for robot development and permits the deletion and reprocessing of functionality. Numerous demonstrations show how this construction technique is broadly applicable, ranging from color-changing robots to warning systems and swimming robots inspired by nature’s chameleons and frogs.

Shark teeth have inspired structures in engineering, as shown by Guo et al. [46], who presented a microneedle patch for smart wound care using advanced materials and biomimetic structures from shark teeth. The patch is made by laser engraving origami and can control drug release and stable adhesion for chronic wound treatment. It also has MXene electronics and microfluidic tubes for biochemical analysis and motion tracking. In vivo tests showed that the patch can promote wound healing in diabetic rats, proving that this bio-inspired design can be applied in the medical field. Although this research was targeted toward wound healing, one cause for concern is how paper origami would react when exposed to environmental conditions commonly present on the skin.

Drawing inspiration from origami-driven devices, Li et al. [47] proposed the architecture of a new fluid-driven origami-inspired artificial muscle (FOAM) that can operate in fluid media by combining a flexible skin and a compressible skeleton. FOAMs can be combined to create multi-DOF systems and produce various motions along multiple axes, including bending, torsion, and contraction. Their ability to function at negative fluid pressures reduces the safety risks relating to fluidic artificial muscles. According to experimental findings, FOAMs can create enormous stresses, contract more than 90% of their starting length, and show peak power densities several times higher than those of natural muscle. FOAMs hold great potential for various applications.

### 3.2. Plantae-Inspired Origami

This section examines the role that complex plant forms and growth patterns have in origami design. As proposed by Dutta et al. [48], there are several degrees of biomimicry, which frequently draw inspiration from specific groups of biological forms depending on their scope and application. Plants exhibit a wide range of structural adaptations for the best use of resources and interactions with the environment, from the graceful unfolding of leaves to the complex blooming flowers. By delving deeper into the biomechanics and developmental processes of plants, design principles that highlight their influences are revealed. Inspired by the kingdom Plantae, these biomimetic origami constructions pave the way for new developments in engineering structures.

Jiao et al. [49] researched origami metamaterials by using a binary digital abstraction, which replicates the growing process of lilies and pushes floriography into the fourth spatiotemporal dimension. The floriography integrates data analysis across time with knowledge from biomimicry, opening up new possibilities for bioinspired intelligent tools and systems. Using theoretical modeling, numerical simulations, and experimental testing, the origami metamaterials’ bistable mechanical response was analyzed, showing its tunability and customizability. The bouquets’ capacity to save and send numbers and letters converted using the American Standard Code for Information Interchange (ASCII) is the most compelling evidence of their information functioning, implying that they are more than just mechanical structures. The selected crease patterns and bistable behavior allow predictable outcomes for such interpretations.

Tang et al. [50] demonstrated how crease patterns can be designed for triangular deployable membranes for space projects via biomimetic folding. Several crease designs, including leaf-in, leaf-out, and orthogonal patterns, were developed and evaluated utilizing performance metrics, including deployment efficiency and linear dimension ratio. These designs were based on biomimetic folding principles. To comprehend how variables influencing folding behavior and deployment efficiency could impact the creases, a parametric study was conducted to determine the most suitable crease patterns and specifications for the triangle deployable membrane used in space missions (Figure 19) [50].

Wong et al. [51] presented a technique for directing large-scale reconfiguration in response to specific water stimuli in soft materials, inspired by the folding of the leaflets in *Mimosa pudica*. Utilizing a Janus bilayer, the system effectively transforms surface energy into kinetic and elastic energies for many-centimeter-scale self-assembly. This mechanism generates exceptional flow rates and velocities while overcoming the drawbacks of previous wicking-based systems. A regime of Mimosa origami with infinite length scaling has considerable potential for applications in biosensors, microfluidics, and water purification. System-able reversible self-assembly makes this method exceedingly adaptable and relevant to various functions in stimuli-responsive matter, artificial muscles, sensors, and other power-independent devices. It also permits the unfolding and regaining of original properties (Figure 20) [51]. While the research to date has been conducted with water, further research could investigate Mimosa origami’s reaction to different fluids and analyze these results.

Yasuda et al. [52] demonstrated an origami structure inspired by natural geometric patterns in the form of a leaf that can move like a Venus flytrap. This allows for highly adjustable, uniform, and non-uniform grasping motions. It was presented as a flexible, autonomous, and self-adaptive robotic operation, showing self-adaptive grasping motion through numerical analysis and experiments. Their research mainly focused on controlling adaptive grasping motion and snap-through behavior by tuning structural parameters, demonstrating adaptive and responsive robotic mechanisms. Notably, leaf-out origami provides researchers with autonomous and versatile grasping motions that warrant the investigation due to their unique kinematics.

Li and Wang [53] studied fluidic origami as adaptive structures inspired by origami folding and plant movements. Their work linked Miura-folded sheets to enable autonomous shape morphing or stiffness adjustment. By filling the cells between the sheets with working fluid, their research opens up new possibilities around linking autonomous motion with internal material properties. They described the structural deformation of multifunctional origami unit cells through a model combining internal fluid volume, folding motion, and material deformation. An experimental case with a 3D printed prototype, shown in Figure 21 [53], demonstrated the feasibility of these implementations.

Similarly, Sun et al. [54] presented soft actuators that respond to different solvent vapors and can move in a similar way to *Dendrobium orchids* and *Bauhinia variegate*. They used the finite element method to study the transient response and recovery actuation parameters and applied Principal Component Analysis (PCA) to the analyte molecules. Their actuator’s origami technique was versatile, and can be applied to grippers, walkers, and snake-like robots. The results showed high addressability and reversibility in mechanical deformation, providing a way to create intelligent actuators with complex motion for various applications (Figure 22) [54]. These three from Yasuda, Li, and Sun show the possibilities of combining origami with adaptive and responsive mechanisms. Yasuda et al. [52] focused on adaptive and self-regulating motion in a leaf-like structure for robotic application, while Li and Wang [53] demonstrated fluidic origami for autonomous shape morphing and stiffness adjustment through internal material properties. Sun et al. [54] used solvent-responsive soft actuators for programmable complex motion and showed versatility in different actuator forms. The common challenge among these studies is to model and control the complex interactions between the material properties and the responsive behavior of the structure so as to obtain precise and reliable performance in real-world applications. Ultimately, origami structures exhibit versatile characteristics that hold the promise of new discoveries in future research.

Poppinga et al. [55] focused on intriguing plant movements and how biomimetics might leverage them. They examined the principles of plant movement and demonstrated how they might serve as an inspiration for the creation of technical systems that move. A variety of diverse principles of plant movements are emulated by original 3D printed hygroscopic shape-changing structures, including snap-through elastic instability actuation, modular aperture architectures, scale-like bending structures, origami-like curved folding kinematic amplification, and motion actuation by edge growth-driven actuation. These intricate origami-style folding, bending, and buckling motions allow for more realistic biomimicry. For example, Sidney et al. [56] studied shape-morphing systems motivated by nastic plant motions and self-folding origami. They presented dynamic conformations such as those of plants through the printing of “composite hydrogel structures” with swelling behavior regulated by the alignment of cellulose fibrils. In order to create the necessary shapes, such alignment patterns are designed within a theoretical framework; this could enable the production of arbitrarily complex three-dimensional morphologies through programming.

The kinematics of a leaf-out origami and its potential to provide a multitransformable structure without the need to modify the crease patterns was examined by Yasuda et al. [57], who investigated several geometrical arrangements of origami by varying the folding and unfolding techniques. Multiple potential energy routes and distinct potential energy values were discovered during transformation when the motions were modeled using linear torsion springs along the crease lines. The origami exhibited bistability, providing snap-through mechanisms as well as being readily deformable for engineering applications, architecture, and space structures. Rigid panels and hinges can be used to take advantage of the mechanics of rigid foldability associated with leaf-out origami, resulting in stiff yet multitransformable structures. Compared to previous foldable structures, the single-degree-of-freedom architecture provides easier control, and as such exhibits potential advantages for a variety of applications.

Pan et al. [58] introduced TransfOrigami microfluidics (TOM), modeled after the nastic movements of plants. TransfOrigami is able to change its morphology in response to external stimuli, mimicking the adaptive reactions of plants to changes in temperature, humidity, and light irradiation by fusing materials that respond to stimuli with foldable geometries. Potential applications for this transformational ability include shape-adaptive flexible electronics, environmentally adaptable photomicroreactors, and dynamic artificial vascular networks. Flexible and stable configurations between flat 2D and 3D can be achieved through origami-inspired structures; however, there is room for improvement in the 4D printing mechanisms, which currently do not support embedding required microtubes (Figure 23) [58].

## 4. DNA Origami and More

The novel idea of DNA origami incorporates nanoscale engineering and molecular self-assembly. Similar to traditional origami that folds paper, DNA origami applies the same concepts to fold nanostructures. Researchers have created methods for folding DNA strands in DNA origami that allow for incredibly exact forms and patterns, creating a whole new field of nanotechnology. Thanks to its programmability and versatility, DNA has become a building block for the creation of nanostructures with uses ranging from medication delivery to nanoelectronics. Examining the traits and properties of DNA origami has led to discoveries that could revolutionize fields such as information technology, materials science, and health. Azulay et al. [59] looked into the analogies between proteins and origami and how origami can explain the folding mechanisms and structural properties of protein folding. By simulating protein folding with origami, researchers can obtain insights into complex folding sequences and misfolding. Marras et al. [60], on the other hand, looked into the programmable nature of DNA origami at the nanoscale to create intricate geometries and reversible motion in DNA origami machines. They showed how flexible DNA origami joints can be designed to move along specific degrees of freedom, such as linear and rotational motion. Evaluating, contrasting, and rearranging nanostructures through DNA origami can provide meaningful results; however, this is a fairly recent topic, with much research pending completion.

On the other hand, Fragasso et al. [61] showed the functional aspect of DNA origami nanopores in lipid membranes (Figure 24) [61] and how they can be used for size-selective diffusion of molecules such as proteins. This application highlights the versatility of DNA origami in biotechnological applications, including drug delivery and biomimetics. Wagenbauer et al. [62] expanded on the practical applications of DNA origami and discussed methods for creating complex structural motifs and performing oligomerization. They also addressed technical challenges with methods such as size-exclusion chromatography and ultracentrifugation to make DNA origami more accessible and useful across many scientific disciplines. All of these studies have in common the use of origami-inspired methods in different scales and materials (protein and DNA). Azulay et al. [59] and Marras et al. [60] used origami to engineer functional structures, but at different scales (macroscale for proteins and nanoscale for DNA). Fragasso et al. [61] and Wagenbauer et al. [62] focused on the practical applications and technical advancements of origami-based technologies in biotechnology and structural design, respectively. Challenges in this area include maintaining structural integrity and functionality across different scales and materials, ensuring precise control over folding and motion dynamics, and integrating these technologies into practical applications such as drug delivery and cellular engineering. Each of these studies shows the interdisciplinary potential of origami-inspired design, from fundamental insights into folding mechanisms to advanced applications in biotechnology and materials science.

A DNA origami technique that allows lengthy DNA sequences to be efficiently folded into various two-dimensional structures was researched by Andersen et al. [63]. Their approach was showcased by building a flexible dolphin-shaped DNA origami structure that can be manipulated and identified using Atomic Force Microscopy (AFM). Their article explored the stability and flexibility of several dolphin tail design configurations, showing how tail attachment sites facilitate dimerization between two dolphin structures to stabilize the flexible tail. Their work offers potential applications in the fields of nanorobotics and nanocantilever technology. Soft DNA origami structures can be a useful tool in creating dynamic nanostructures tailored to specific applications (Figure 25) [63]. Protein functionality is determined by the protein’s folded state and structure; therefore, learning how to fold nanostructures can open up new doors in the biomedical engineering field and research into specific diseases.

In an article published in ACS Nano, Suzuki et al. [64] demonstrated how lipid-bilayer-anchored DNA origami structures can be combined into specific structures that resemble the lively congregation of protein clusters necessary for cellular membrane deformation, highlighting a significant advancement in DNA nanotechnology. This opens the door for DNA nanostructures to be applied in artificial cells in the future, with applications spanning biotechnology and nanomedicine. The reversibility of assembly reactions and speed are two main challenges in these assemblies. It is crucial to understand how DNA origami structures behave and interact with biological membranes in order to build DNA-based artificial cellular components in the future. Because DNA molecules are extremely programmable, there is a plethora of fascinating options for replicating cellular activities with DNA nanostructures, allowing for innovative study in this area (Figure 26) [64].

Lapenta et al. [65] reviewed the field of artificial, chemical-based, and biological structures opened up by the development of modular protein assemblies based on coiled-coil (CC) peptide secondary structure elements. By applying the rules governing the specificity of CC pairing, these assemblies provide an incredibly flexible framework for creating functional nanostructures resembling DNA modules. These modular CC-based protein structures can assemble into a variety of topologies, including coiled-coil protein origami, by using orthogonal construction modules that couple and assemble only with predetermined partners. Building coiled-coil protein origamis that convene in vivo without evident harm has been made possible by recent advancements in CC modules and design principles. These compounds have the potential to be used biomedically and therapeutically for purposes such as medication carriage. Additionally, Buchberger et al. [66] highlighted the use of orthogonal coiled-coil peptides in the synthesis of bioactive fibronectin domain proteins with DNA nanostructures. Their method allows for the creation of a DNA nanofiber arrangement in a single dimension with intermittent bioactive fibronectin domains by genetically fusing peptides complementary to the DNA origami cuboid to bioactive fibronectin domains. This technique is the first to show how the coiled-coil motif’s bioactivity contributes to the creation of hybrid self-assembling structures. This process produces nanofibers that are more active than monomeric proteins and serve as useful scaffolds for cell attachment and dissemination.

In addition to discussing the structural integrity, denaturation, and deterioration of these nanostructures under various environmental conditions pertinent to biologically related fields and materials science, Ramakrishnan et al. [67] reviewed the sensitivity of DNA origami to temperature, cation concentration, and nuclease attack. Protective techniques such as coatings and chemical alterations are necessary when using DNA origami. In biological contexts, DNA origami rapidly degrades, although it remains robust under exceptionally difficult conditions such as high temperatures and denaturing chemicals. To control stability and nuclease resistance, DNA origami design and superstructure are emphasized. This establishes a foundation for future study into carefully crafted DNA origami for varying applications. Although the DNA origami structures can be modified and adjusted, there remains cause for concern about how stable these final structures are and how functional they will remain over time.

## 5. Discussion

There are several hurdles in origami engineering that will determine its future outlook; these include maintaining structure and function across scales and materials, control over folding dynamics and motion mechanisms, integrating origami into biomedical devices and robotics, and fabricating origami structures. In addition, solving problems related to scalability, durability, adaptability to different environments, and technical barriers in manufacturing such as precision folding, material compatibility, and structural stability will be key to unleashing the promise of origami in fields from nanotechnology and biomedicine to aerospace and beyond.

## 6. Conclusions

The goal of the paper has been to explore the fascinating field of origami engineering, which draws inspiration from both the inventive field of DNA origami and the two major kingdoms of life, Animalia and Plantae. This field reveals a wealth of opportunities for engineering applications through the methodical investigation of bioinspired designs and systems. Origami structures serve as adaptable answers to a wide range of real-world problems, with solutions ranging from shape-morphing microfluidic systems that significantly enhance environmental responsiveness to biomimetic grippers that enhance industrial automation. The applications of these technologies could be enormous and revolutionary. By pushing the boundaries of origami engineering and utilizing the grace and efficiency of nature’s systems, research in this area is reshaping industries, improving human welfare, and advancing scientific understanding.

## Figures and Tables

**Figure 1 biomimetics-09-00600-f001:**
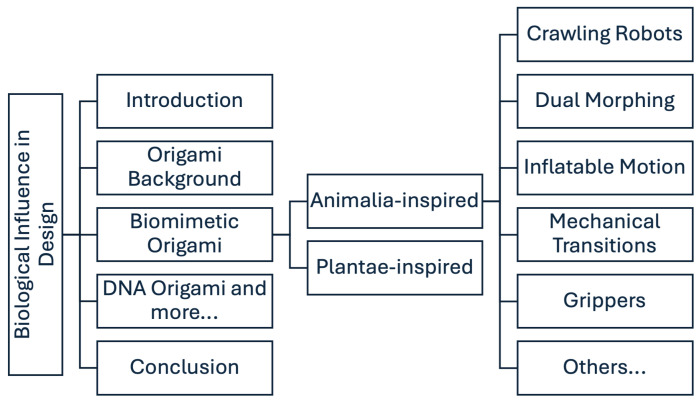
Outline and structure of the content covered in this review paper.

**Figure 2 biomimetics-09-00600-f002:**
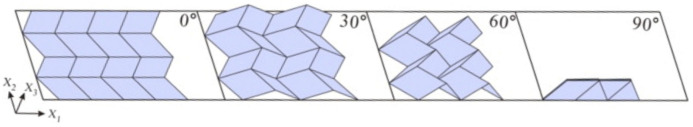
Range of motion for the Miura Origami pattern at fold angles of 0°, 30°, 60°, and 90° [6].

**Figure 3 biomimetics-09-00600-f003:**
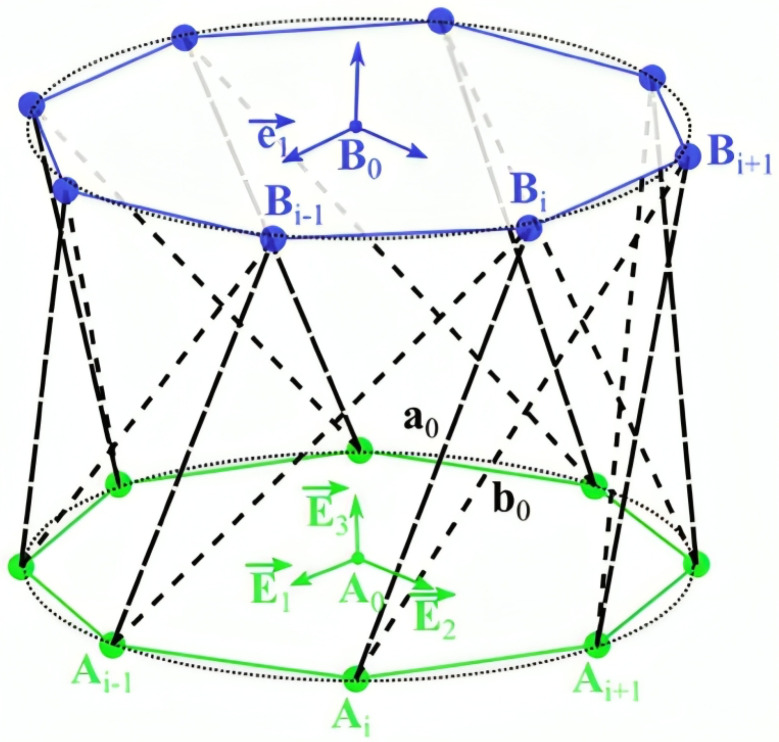
Range of motion for the deployed Kresling Origami pattern, where the top blue lines converge with the bottom green lines at the folded state [7].

**Figure 4 biomimetics-09-00600-f004:**
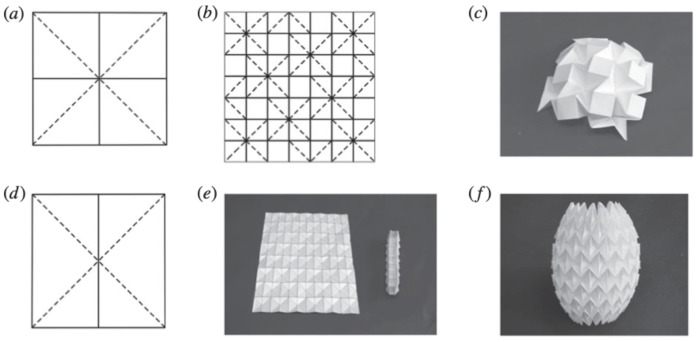
Waterbomb pattern [14]: (**a**) eight-crease waterbomb, (**b**) waterbomb Resch tessellation, (**c**) semi-folded Resch tessellation, (**d**) six-crease waterbomb, (**e**) folded and unfolded six-crease waterbomb, (**f**) cylindrical waterbomb fold.

**Figure 5 biomimetics-09-00600-f005:**
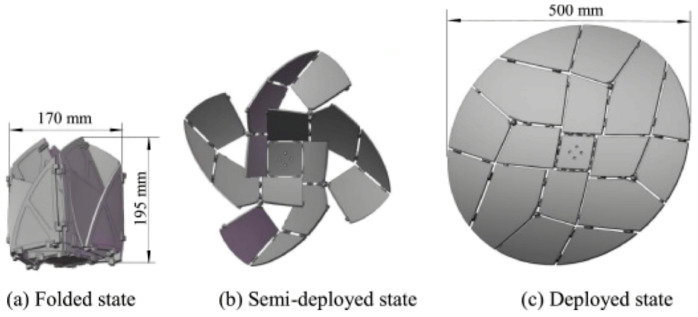
Flasher pattern [15].

**Figure 6 biomimetics-09-00600-f006:**
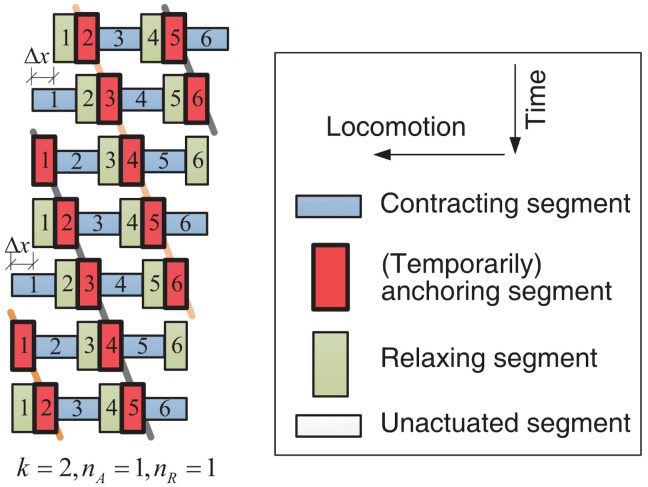
Example of earthworm-like locomotion gait [17].

**Figure 7 biomimetics-09-00600-f007:**
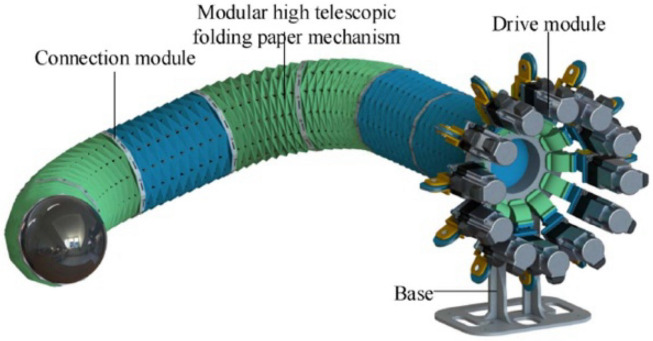
Continuum robot with four modules following the Yoshimura pattern, intended to achieve high elongation [20].

**Figure 8 biomimetics-09-00600-f008:**
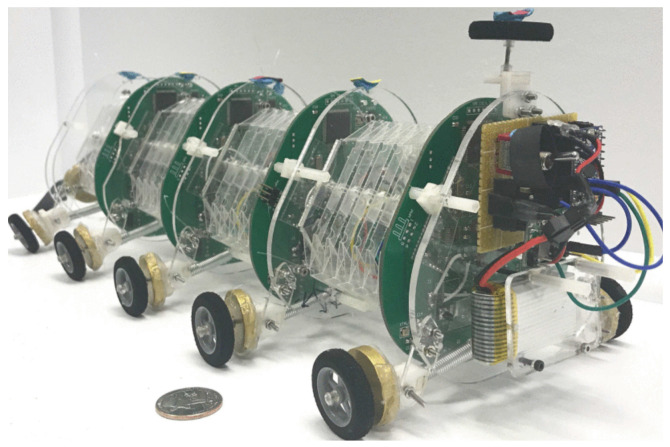
Origami Robotic Snake (OriSnake) robot employing a 3D Yoshimura origami pattern. Its structural design allows it to withstand torsional deformation while bending, contracting, and expanding [23].

**Figure 9 biomimetics-09-00600-f009:**
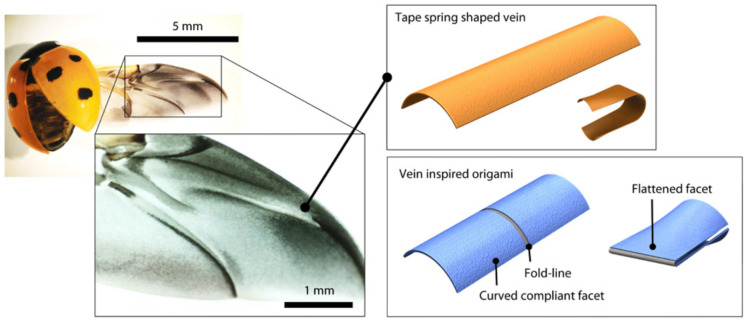
Ladybird beetle-inspired compliant origami [26].

**Figure 10 biomimetics-09-00600-f010:**
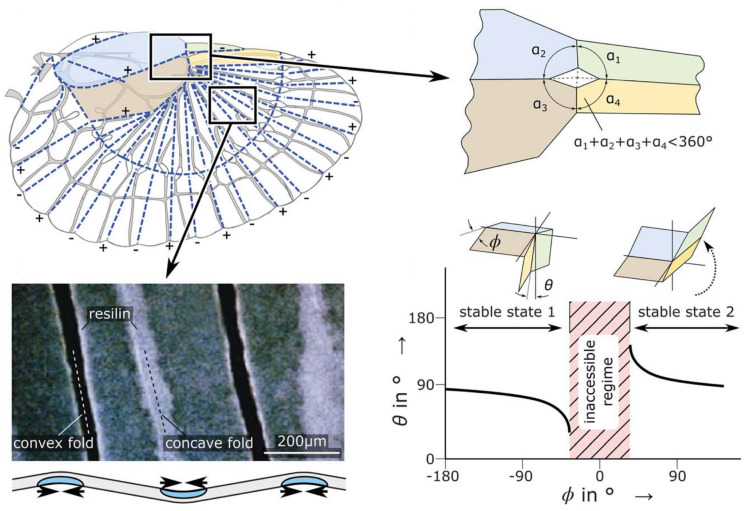
The earwig wing’s natural multifunctional programmable folding [27].

**Figure 11 biomimetics-09-00600-f011:**
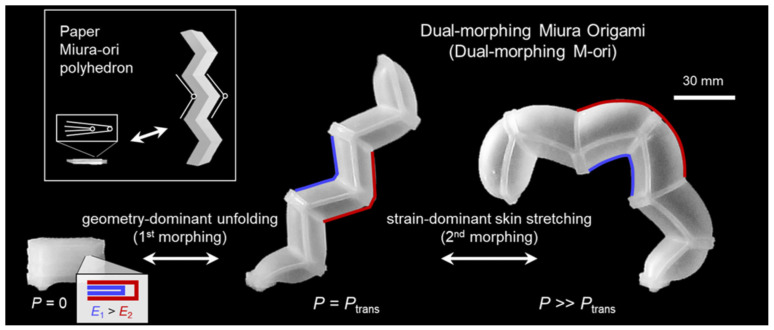
Six-module Miura fold origami inspired by unfolding and inflating behavior of the pelican eel [30].

**Figure 12 biomimetics-09-00600-f012:**
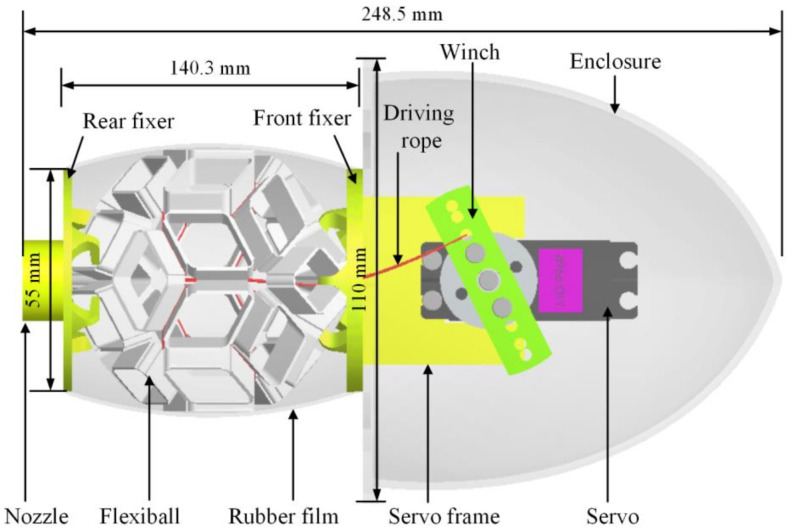
Triacontahedral flexible origami ball with a cavity; the volume of the cavity changes as the ball shrinks or expands, generating a jellyfish-like motion [32].

**Figure 13 biomimetics-09-00600-f013:**
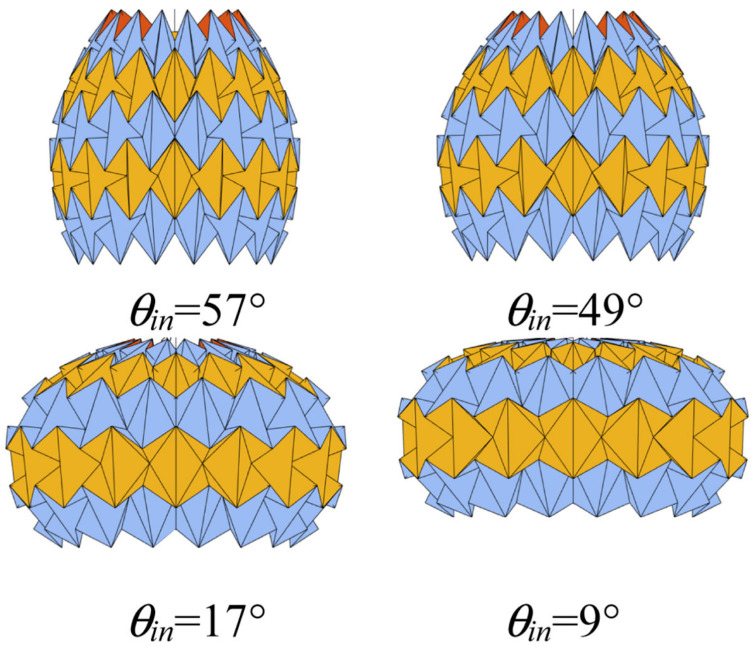
Waterbomb origami structure imitating the motion principles of jellyfish through jetting and rowing propulsion, achieving significant volume variations in different deployment configurations [34].

**Figure 14 biomimetics-09-00600-f014:**
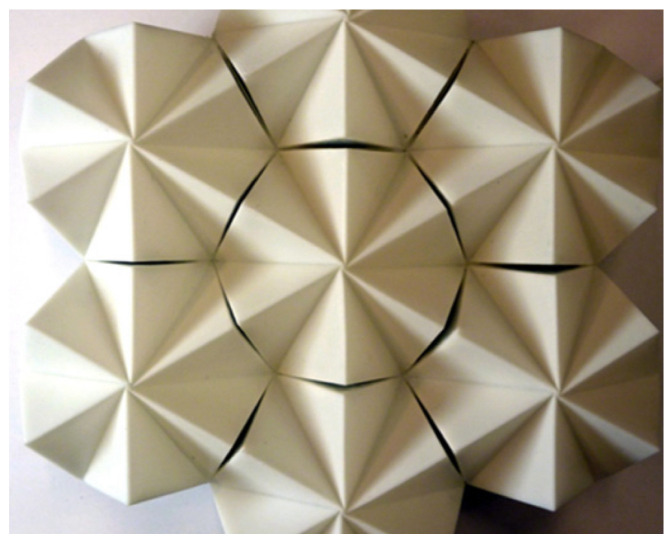
Upper skin morphing cell prototype inspired by cephalopod papillae [35].

**Figure 15 biomimetics-09-00600-f015:**
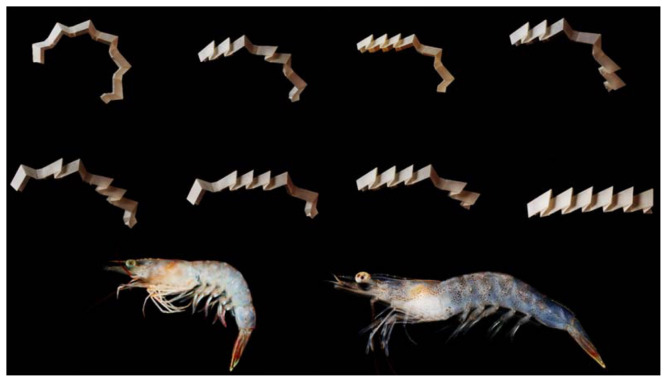
Multistable shrimp pattern tessellation [36].

**Figure 16 biomimetics-09-00600-f016:**
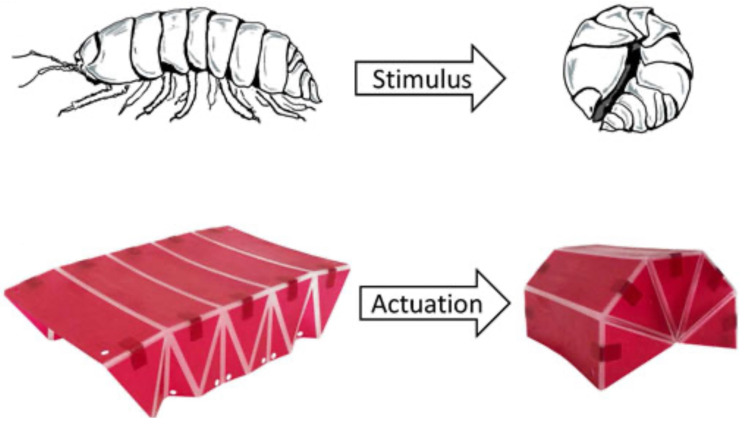
Rolling of a real pill bug (**top**) vs. origami pill bug structure (**bottom**) [38].

**Figure 17 biomimetics-09-00600-f017:**
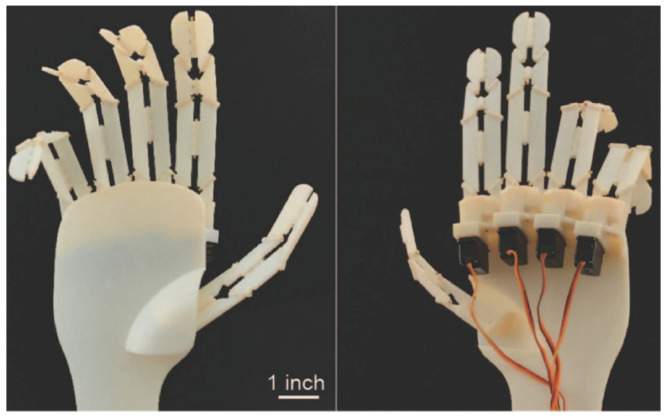
Biomimetic origami hand that can perform intricate movements through incorporation of Miura origami strings [39].

**Figure 18 biomimetics-09-00600-f018:**
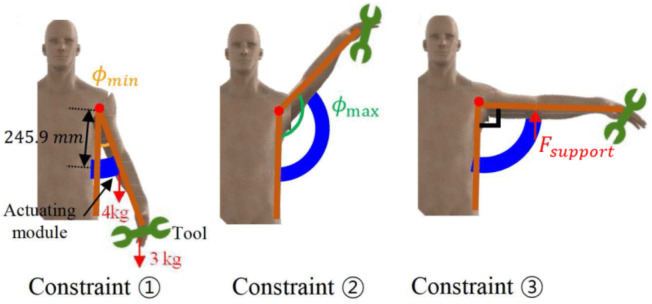
Origami-structured support for upper human limb schematic designed in the Yoshimura pattern (blue) to allow for bending motions by fixing one side of the module [42].

**Figure 19 biomimetics-09-00600-f019:**
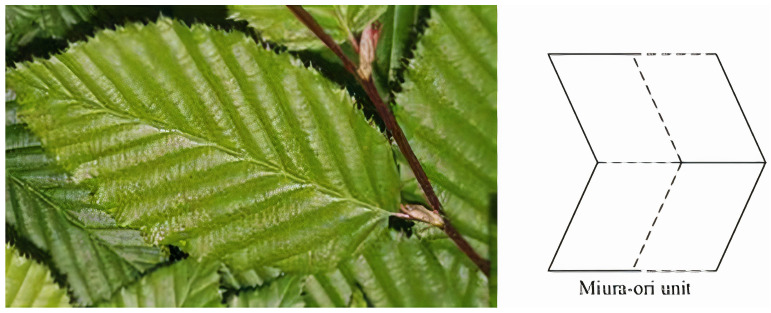
New folding patterns have been inspired by the way in which natural structures fold as a result of millions of years of adaptation to life and constantly shifting environments [50].

**Figure 20 biomimetics-09-00600-f020:**
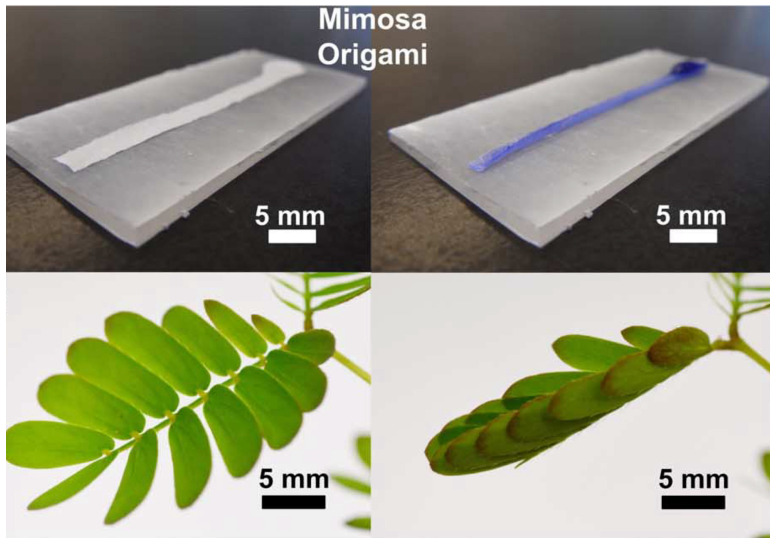
Directional self-organization via Mimosa origami when exposed to water stimuli [51].

**Figure 21 biomimetics-09-00600-f021:**
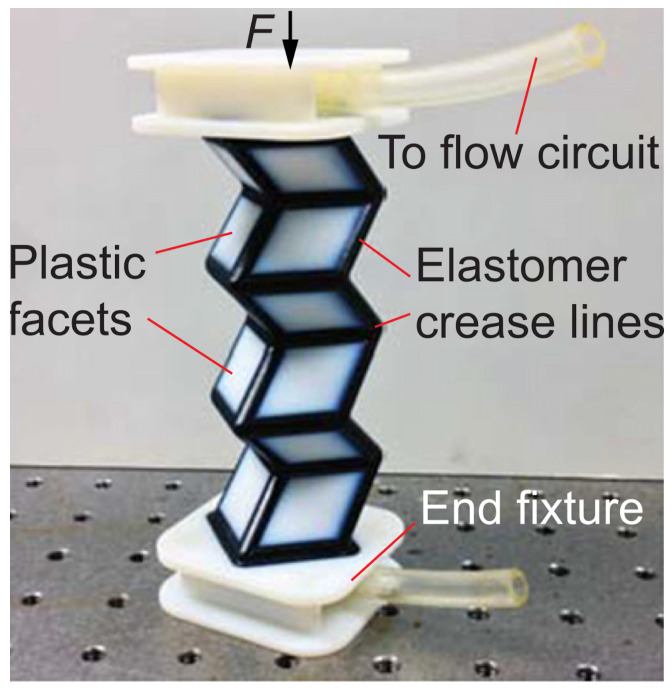
Concept of autonomous shape-morphing Miura structure based on total incoming fluids from the “flow circuit” and exiting fluids through the “end fixture” [53].

**Figure 22 biomimetics-09-00600-f022:**
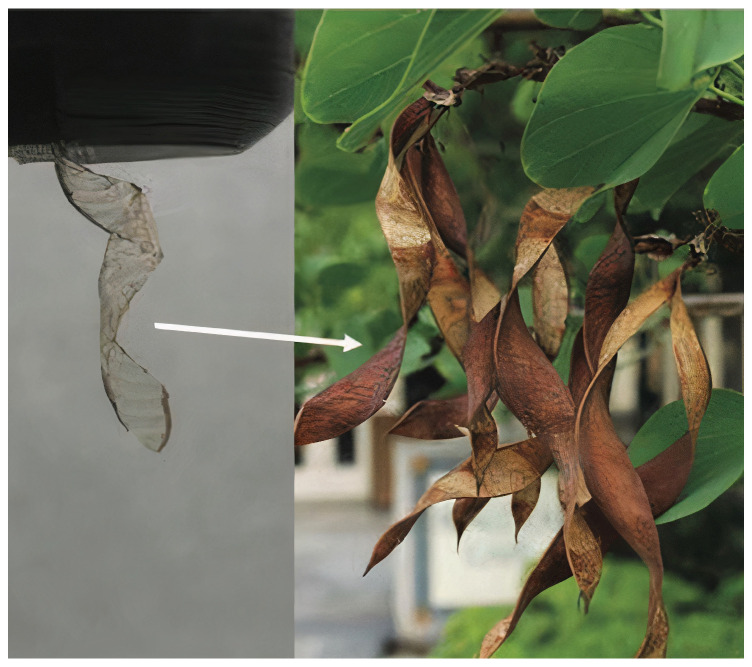
One-side patterned soft actuator exposed to *n*-pentane vapors [54].

**Figure 23 biomimetics-09-00600-f023:**
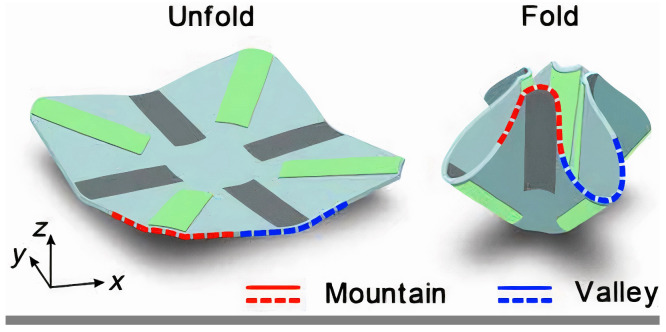
TransfOrigami folding concept in response to external temperature and humidity [58].

**Figure 24 biomimetics-09-00600-f024:**
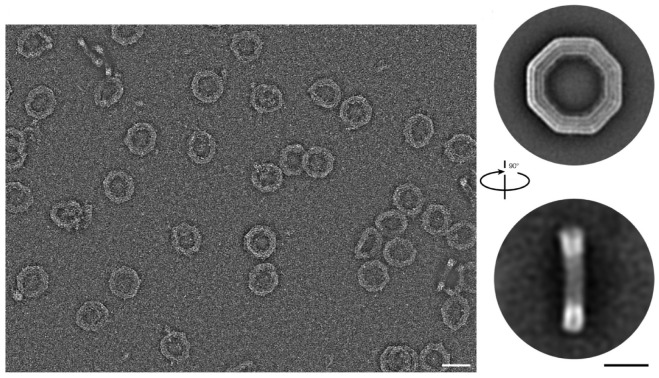
Negative-stained DNA origami pore sample [61].

**Figure 25 biomimetics-09-00600-f025:**
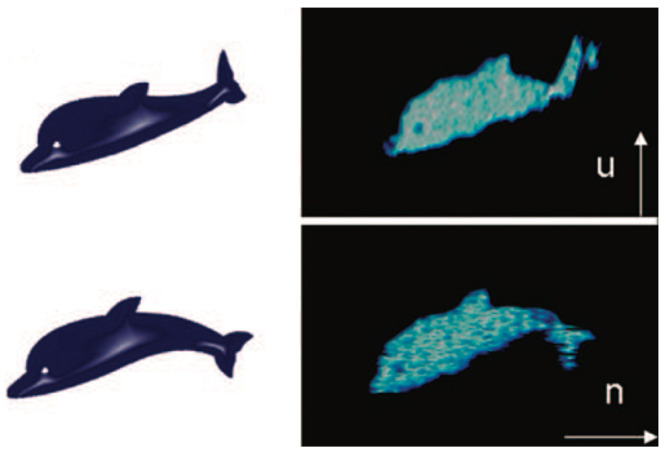
Study of dolphin tail flexibility with DNA origami [63].

**Figure 26 biomimetics-09-00600-f026:**
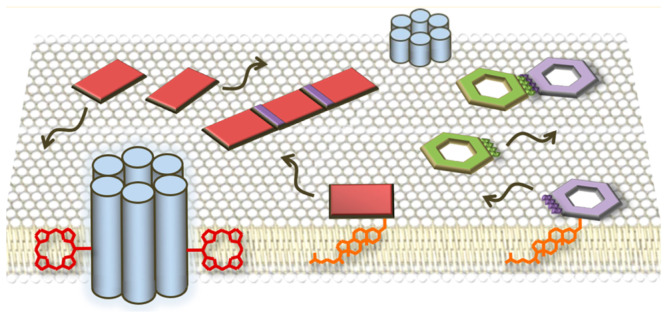
Controlled gathering of DNA origami nanostructures [64].

## Data Availability

No new data were created or analyzed in this study. Data sharing is not applicable to this article.

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
