# Peer review of "Biomimetic Origami: A Biological Influence in Design"

_biomimetics, 2024, doi:10.3390/biomimetics9100600_

Round 1
Reviewer 1 Report (New Reviewer)
Comments and Suggestions for Authors
Overall, the paper is interesting and well written. The analysis and discussion of results was thorough. The font size on many of the figures is too small and the white and black circles promised in Figure 9 were not detectable. Possible add reference to the second to the last sentence in the first paragraph.
Author Response
Please see the attachment.

Reviewer 2 Report (New Reviewer)
Comments and Suggestions for Authors
A review of the current state of biologically-inspired origami structures is presented. Biomimetic designs that resemble intricate patterns and functionalities occurring in nature are analyzed. The engineering structures inspired by morphological features and its behaviors are explored. The implications in realms like sustainability, biomaterials development, and bioinspired robotics are discussed. But this is not a very successful review paper. Much of the recent literature are not investigated or mentioned. The review should be extended to include some recently published paper in top-tier journals. Research works are briefly listed, but few comparative studies are presented. Based on the above comments, the current version of the paper is not suitable for publication in this journal.
Author Response
Please see the attachment.

Reviewer 3 Report (New Reviewer)
Comments and Suggestions for Authors
I think the paper is well organized and showcases a plethora of applications of bioinspired Origami and inspiration for different design strategies. The content will be helpful for the designers and engineers looking to gain an understanding of how the Origami mechanisms can be used. However, the general tone and summary of applications and technological impact described in the paper are hard to follow for the novices, who might not be well versed in bistability, multi-transformability, nonlinear stiffness, etc.
Please find below my comments and suggestions:
- This is a review paper to highlight the attractiveness of biomimetic Origami. I think it should add a few more relevant references on each topic they are describing. This will help newcomers to the field gauge the state-of-the-art of this technology.
- In the paragraph about the Kresling fold the authors should mention the bistability of Kresling Origami, which is one of its biggest strengths.
- Some additional examples of rotationally symmetric Origami patterns such as the Flasher Origami Pattern would round out the description of commonly used. origami patterns.
- A brief paragraph mentioning variable and negative stiffness, multi-transformability, Auxetic property, and multi-stability through a few examples will enrich the paper even more. It will also highlight why engineers prefer to use Origami for these applications.
- The figures in the paper are hard to understand and would benefit from adding a general title followed by a brief explanation of what is shown in the figure. Also, organizing the figures near the relevant passage would help the reader make connections easily.
- The conclusion of the review article combines the future outlook with the paper summary. Maybe adding a separate paragraph to describe the future outlook/importance of biomimetic Origami is helpful.
General Comments:
- The general flow of the paper is good, but the individual reference summaries are hard to follow.
- Simplifying the technical jargon and sentences would help the reader.
- Avoid/minimize using passive voice sentences wherever possible, it makes them hard to follow.
Specific comments per line:
- Line 69 should be: of Koryo Miura
- Line 209 - 211: please rewrite confusing and very long sentences that are hard to make sense of.
- Edit the description in Figure 13 to make it self-explanatory by adding the Origami pattern being described, its application, and its mechanism features. This comment also applies to many other figures.
- Line 358: what are “obedient living entities”?
- Line 377: Instead of Others… maybe replace with Other Applications
- Line 442: Use “different” instead of “variable”
- Figure 21 text: is very inconsistent and doesn’t explain what is being shown in the figure
- Figure 23 text: is very inconsistent and doesn’t explain what is being shown in the figure
Round 2
Reviewer 2 Report (New Reviewer)
Comments and Suggestions for Authors
Accept in present form.
This manuscript is a resubmission of an earlier submission. The following is a list of the peer review reports and author responses from that submission.
Round 1
Reviewer 1 Report
Comments and Suggestions for Authors
Upon further examination, the reviewer concluded that the authors' modifications were sufficient to allay my worries. I suggest that the article get published.